# Distribution System State Estimation Using Hybrid Traditional and Advanced Measurements for Grid Modernization

Sepideh Radhoush *, Trevor Vannoy, Kaveen Liyanage, Bradley M. Whitaker and Hashem Nehrir

Electrical and Computer Engineering Department, Montana State University, Bozeman, MT 59717, USA; trevorvannoy@montana.edu (T.V.); kaveen.liyanage@student.montana.edu (K.L.); bradley.whitaker1@montana.edu (B.M.W.); hnehrir@montana.edu (H.N.)
* Correspondence: sepideh.radhoush@student.montana.edu

**Abstract:** Distribution System State Estimation (DSSE) techniques have been introduced to monitor and control Active Distribution Networks (ADNs). DSSE calculations are commonly performed using both conventional measurements and pseudo-measurements. Conventional measurements are typically asynchronous and have low update rates, thus leading to inaccurate DSSE results for dynamically changing ADNs. Because of this, smart measurement devices, which are synchronous at high frame rates, have recently been introduced to enhance the monitoring and control of ADNs in modern power networks. However, replacing all traditional measurement devices with smart measurements is not feasible over a short time. Thus, an essential part of the grid modernization process is to use both traditional and advanced measurements to improve DSSE results. In this paper, a new method is proposed to hybridize traditional and advanced measurements using an online machine learning model. In this work, we assume that an ADN has been monitored using traditional measurements and the Weighted Least Square (WLS) method to obtain DSSE results, and the voltage magnitude and phase angle at each bus are considered as state vectors. After a period of time, a network is modified by the installation of advanced measurement devices, such as Phasor Measurement Units (PMUs), to facilitate ADN monitoring and control with a desired performance. Our work proposes a method for taking advantage of all available measurements to improve DSSE results. First, a machine-learning-based regression model was trained from DSSE results obtained using only the traditional measurements available before the installation of smart measurement devices. After smart measurement devices were added to the network, the model predicted traditional measurements when those measurements were not available to enable synchronization between the traditional and smart sensors, despite their different refresh rates. We show that the regression model had improved performance under the condition that it continued to be updated regularly as more data were collected from the measurement devices. In this way, the training model became robust and improved the DSSE performance, even in the presence of more Distributed Generations (DGs). The results of the proposed method were compared to traditional measurements incorporated into the DSSE calculation using a sample-and-hold technique. We present the DSSE results in terms of Mean Absolute Error (MAE) and Root Mean Square Error (RMSE) values for all approaches. The effectiveness of the proposed method was validated using two case studies in the presence of DGs: one using a modified IEEE 33-bus distribution system that considered loads and DGs based on a Monte Carlo simulation and the other using a modified IEEE 69-bus system that considered actual data for loads and DGs. The DSSE results illustrate that the proposed method is better than the sample-and-hold method.

**Keywords:** distribution system state estimation; weighted least square; SCADA measurements; PMU measurements; grid modernization; multioutput regression

# 1. Introduction

Due to the scarcity of measurement devices in Distribution Networks (DNs), Distribution System State Estimation (DSSE) calculation is challenging [1]. In DNs, the lack of measurements causes issues with observability and the ability to perform DSSE computations. Generating pseudo-measurements, which are acquired from load forecasting based on Advanced Metering Infrastructure (AMI) measurements, is a common way to address these issues [2,3]. However, the quality of the pseudo-measurements is low in both speed and accuracy [4]. Conventional measurements from Control and Data Acquisition (SCADA) measurements, along with pseudo-measurements, are used to perform State Estimation (SE) calculations using low sample rates [5,6], and SCADA measurements are asynchronous and available with a long interval time [7]. In addition, the conventional DSSE results are not precise enough, since SCADA measurements have low accuracy and are much higher than the pseudo-measurement [5,8]. Moreover, new features, including DGs, two-way communications, electric vehicles, variable loads, etc., will increase the need for accurate DSSE calculations [9,10]. Therefore, installing smart measurement devices, such as Phasor Measurement Units (PMUs) and micro PMUs ($\mu$PMUs), currently involve grid modernization projects to improve DSSE results [11,12]. PMUs are extensively used in Transmission Networks (TNs) [13]. However, because DNs have different characteristics from TNs, PMUs are not as commonly used in DNs [14]. Some DN characteristics are:

1. DNs typically have radial or weakly meshed topology.
2. The placement of measurement tools is not feasible at all buses due to economic constraints and DN configuration with multiple buses.
3. Radial DNs have a high resistance-to-reactance ratio ($r/x$).

Since PMU measurements are available in a synchronous form and can achieve high sampling rates, the incorporation of PMU measurements along with traditional measurements into SE calculations improves DSSE results, as well as increases the observability, resiliency, and reliability levels of DNs [15–17]. However, replacing conventional measurement devices with advanced tools is not possible over a short period of time; it may take several years to modernize a DN. Because of the presence of both traditional and advanced measurement data during the modernization process, managing, controlling, and performing SE calculations can be challenging [18].

## 1.1. Literature Review

One of the important issues in the grid modernization process is how different measurements can be incorporated correctly in a control center to perform SE calculations accurately during high-speed operations [19,20]. Inconsistent reporting rates of measurement devices, as well as different types of measurement information with various precision levels, are two of the most critical problems for performing hybrid SE calculations from traditional and advanced measurement devices [21]. PMUs commonly provide synchrophasor measurements at a rate of 30 samples per second as compared to SCADA measurements, which update every 2–6 s [22,23]. Hybrid SE methods have been well established in TNs to integrate and make use of the potential benefits of traditional and advanced measurements [24,25]. Different hybrid transmission state estimation methods to combine traditional and advanced measurements are discussed in [26]. Hybrid methods will be helpful during the power grid modernization development while replacing all-traditional with all-advanced measurement devices.

Machine learning techniques—including supervised, unsupervised, and reinforcement approaches [27,28]—are applied widely in DNs for different purposes, such as generating pseudo-measurements [29,30], SE calculation [31,32], False Data Injection Attacks (FDIAs) detection [33,34], reliability and resiliency analysis, etc. [35,36]. In [37], a Deep Neural Network (DNN) approach was applied for topology and SE in DNs. Machine learning methods have also been developed to improve hybrid traditional and smart measurement data for DN modernization. In [38], different attributes of PMU and SCADA measurements are determined based on data accuracy, time scale, and refreshing frequency. Then, a

specific solution is created based on the differences between measurement properties. In [9], in order to include both SCADA and PMU measurements for DSSE calculation, the state estimator switched between Weighted Least Square (WLS) and Weighted Least Absolute Value (WLAV) calculations. In [39], a new hybrid SE method based on Particle Swarm Optimization (PSO) [40] and Artificial Neural Network (ANN) approaches was proposed to combine the different measurements. In [41], nonlinear static state estimation, linear static state estimation, and linear dynamic state estimation were applied to hybrid SCADA, PMU, and AMI measurements in ADNs. In [42], a new hybrid state estimator considering a distributed state estimation method was proposed to use SCADA and $\mu$PMU measurements in an unbalanced DN. In [43], improved robust and linear state estimators were applied to respond to different frequency levels on measurement data from a SCADA and PMU for medium voltage DNs. In [44], a hybrid simultaneous state estimator based on the WLS method was proposed to incorporate both SCADA and PMU measurements. In [45], a hybrid SE method based on WLAV estimator and Deep Neural Network (DNN) methods was proposed to manage the unsynchronized AMI and SCADA measurements and to monitor ADNs. In [46], SCADA, $\mu$PMU, and Smart Meter (SM) data were used for unbalanced DNs, where Kalman smoother and Expectation Maximization (EM) methods were used to manage the missing measurements and combine measurements in different time samples, respectively. In [47], a static state estimator was used to perform DSSE calculations using both SCADA and PMU measurements. When SCADA measurements were not available, a fast state reconstruction algorithm was applied using only PMU and pseudo-measurements. In [48], a Spatio-Temporal Estimation Generative Adversarial Network (ST-EGAN) was proposed to generate pseudo-measurements to perform DSSE calculations in high resolution when SCADA and PMU measurements were not available simultaneously. In [49], an improved sequential state estimation method was proposed to take advantage of AMI and $\mu$PMUs measurements to address the asynchronization issue for performing DSEE calculations.

### 1.2. Contribution

As indicated in the previous paragraph, a number of previous studies have focused on using traditional and smart measurements to improve DSSE results. However, they generally have switched between two distinct SE methods considering the types of available measurements. Moreover, the most recent studies in the hybridization of traditional and smart measurements assume that both traditional and smart measurements are available from the initial distribution network operation. As a result, the system cannot be effectively modeled the moment smart measurements are available. In addition, any model training must be redone every time a new sensor is added to the network. This makes their methods non-applicable for grid modernization development; in the real world, advanced measurements are not available from the initial operation of a DN; rather, they are added gradually while considering the requirements and needs of DNs.

In contrast, this paper assumes that only SCADA measurements are available from the beginning operation of a DN and that PMU devices are added later as part of a grid modernization procedure.

The main contributions of this paper are summarized as follows:

1. SCADA measurements and pseudo-measurements were used to perform DSSE calculations based on the WLS method before the installation of PMU devices and additional DGs in ADNs.
2. A multi-output regression model was trained using DSSE results from SCADA and pseudo-measurements.
3. This model predicted SCADA measurements after smart measurement devices were added to the network, when these measurements were not available, to enable synchronization between the traditional and smart sensors despite their different refresh rates.
4. As more data was collected from the measurement devices, the regression model performance improved under the condition that the model continued to be updated

regularly. Thus, the training model became robust and improved DSSE performance even in the presence of more Distributed Generations (DGs) or other dynamic changes.

5. The results of the proposed method were compared to traditional measurements incorporated into the DSSE calculation using a sample-and-hold technique. This analysis was applied to two case studies: (1) an IEEE 33-bus case study using loads and generation from a Monte Carlo simulation and (2) an IEEE 69-bus case study using loads and generation from actual data.

It is worthwhile to mention that, in this work, we aimed to address the absence of SCADA measurements between PMU measurements using a regression model instead of switching between two different DSSE methods. The regression model was updated as more DSSE results were obtained to improve its accuracy. Note that we did not consider execution time in this study.

## 2. Traditional and Smart Measurement Devices

The penetration of DGs, EVs, and variable loads bring complex interaction characteristics with high uncertainties that makes the operation, monitoring, and management of DNs more challenging [50,51]. Control and Data Acquisition (SCADA) measurement systems are currently applied to monitor and control DNs [52]. The information from Remote Terminal Units (RTUs), which are conventional measurement devices, are installed at various locations of DNs, which are recorded through SCADA [53]. Measurement data are unsynchronized voltage and branch current magnitudes, real and reactive power injections, and flows within a lowest refresh interval time [54]. These measurements, which are available using SCADA systems, fail to capture the dynamic behavior of ADNs [55]; therefore, PMUs are introduced to facilitate a suitable and reliable monitoring of ADNs in terms of higher precision and speed [56]. Phasor Measurement Units (PMUs) are advanced measurement devices in power grids that provide an accurate and synchronized voltage bus and branch current phasor measurement [57]. PMUs have been widely used for wide-area monitoring, protection, and the control of TNs [58]. However, PMUs have not been deployed widely in ADNs due to economic constraints as well as structure of the DNs [59,60]. DN monitoring will be enhanced by incorporating more PMUs in an ongoing modernization effort for ADNs due to the following reasons [61,62]:

➢ The linear mathematical SE calculation will be formulated, since state variables could be measured by PMUs.

➢ PMU measurements are available with a high precision (0.1% and 0.01 rad for magnitudes and phase angle, respectively); thus SE computations can be performed using PMU measurements with high-speed processing and accuracy.

➢ Tracking the dynamic behaviors of ADNs could be captured using high-resolution PMU data in DSSE calculations.

SE results will be enhanced by including different measurement data from multiple measuring instruments due to redundancy [60]. However, measuring information integration from different sources is not straightforward for two main reasons [63]:

1. The absence of unique coordination time among different measurements leads to time-skew problems, and datasets might not be established in an exact time that is representative in a control center. Time inconsistency might also result from the effect of the communication delays of monitoring devices [64].

2. The precision weights of measured quantities from SCADA and PMU measurements are different; therefore, current SE software should be altered or corrected to make all measurement values applicable in SE algorithms [65].

The SCADA and PMU measurements, as well as their refresh samples, are shown in Figure 1a,b respectively.

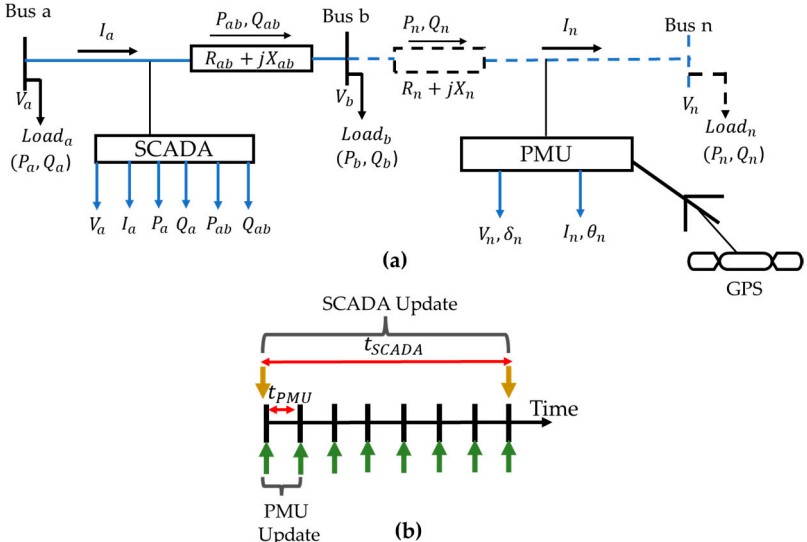

**Figure 1.** (**a**) SCADA and PMU measurements in a power network; (**b**) SCADA and PMU measurement refresh rates.

## 3. Distribution System State Estimation

SE calculation is the function of the Energy Management System (EMS) in a power grid. SE calculations provide a way to monitor real-time power grid operation and develop more power grid analysis, including power flow calculations, power dispatching, observability analysis, bad-data detection, reliability assessment, etc. Transmission state estimation techniques have been established, but these techniques cannot be applied directly for DNs due to the differences between transmission and distribution grids. New DSSE strategies should be designed according to the needs and requirements of DNs. The first DSSE technique was established when the SCADA system was introduced in 1990 [66]. Traditional SE techniques are commonly performed based on conventional measurements, which are commonly obtained from SCADA systems [67]. However, since SCADA measurements are not enough to acquire SE calculations, pseudo-measurements are considered along with traditional measurements to make a system observable [3]. In order to calculate state estimation variables, the WLS method is generally applied to both TNs and DNs [66]. By using WLS, the state vector can be defined based on node voltage [68,69] or branch current variables [70,71]. When state variables are calculated, other network parameters can also be determined.

In node voltage definition, bus voltage phase angle, as well as voltage magnitude, are considered as a state vector [72]: $x = [\theta, |V|]$.

When only conventional measurements are available, and no measurement devices are placed at the slack bus, the state vector is defined in polar coordination as follows:

$$x = [\theta_2, \cdots, \theta_N, V_1, \cdots, V_N], \tag{1}$$

where, $N$ is the number of buses. The slack bus is considered as a reference: $\theta_1 = 0$. When a PMU is located at the slack bus, the phase angle at the slack bus is not zero, and it should be involved in the state vector. The state vector is represented as:

$$x = [\theta_1, \cdots, \theta_N, V_1, \cdots, V_N]. \tag{2}$$

The relationship between the state variables and the measurement data is called the measurement function, which is represented as:

$$z = h(x) + e, \tag{3}$$

where $z = [z_1, \cdots, z_M]^T$ is the measurement vector, and $M$ is the number of measurements. Bus voltage magnitudes, phase angles, branch current magnitudes, and phases, as well as active and reactive power flows and injections, can be involved in the measurement vector.

For completeness of the paper, we include the equations for standard active ($P_i$) and reactive ($Q_i$) power injections at bus $i$:

$$P_i = V_i \sum_{j=0}^{N} V_j \left( G_{ij} cos\theta_{ij} + B_{ij} sin\theta_{ij} \right) \tag{4}$$

$$Q_i = V_i \sum_{j=0}^{N} V_j \left( G_{ij} sin\theta_{ij} - B_{ij} cos\theta_{ij} \right). \tag{5}$$

$G_{ij}$ and $B_{ij}$ are the real and imaginary part, respectively, in a nodal admittance matrix element $Y_{ij}$, and $\theta_{ij} = \theta_i - \theta_j$ is the standing phase angle difference between buses $i$ and $j$.

The active ($P_{ij}$) and reactive ($Q_{ij}$) power flows from bus $i$ to bus $j$ are expressed as:

$$P_{ij} = V_i V_j \left( G_{ij} cos\theta_{ij} + B_{ij} sin\theta_{ij} \right) - G_{ij} V_i^2 \tag{6}$$

$$Q_{ij} = V_i V_j \left( G_{ij} sin\theta_{ij} + B_{ij} cos\theta_{ij} \right) + B_{ij} V_i^2. \tag{7}$$

The relationship between the measurement vector and the state variable $x = [x_1, \cdots, x_N]^T$ ($N$ is the number of state variables) is shown by $h = [h_1, \cdots, h_M]^T$. $h$ is the list of measurement functions, which is commonly nonlinear. $e = [e_1, \cdots, e_M]^T$ is the vector measurement error, which follows the Gaussian distribution $e \sim N(0, W)$, where $W$ is the covariance matrix of the measurement errors $W = diag\{\sigma_{z_1}^2, \sigma_{z_2}^2, \cdots, \sigma_{z_m}^2\}$ and the zero mean. The WLS technique is commonly applied to solve an SE problem. In this method, state vectors are obtained by minimizing the sum of the residual squares as follows.

$$\hat{x} = \arg \min_x J(x)$$
$$= \arg \min_x \left( (z - h(x))^T W^{-1} (z - h(x)) \right). \tag{8}$$

the iterative Gauss–Newton numerical method is applied to solve the optimization problem (8) with objective function $J(x)$.

$$[G(x^k)]\Delta x^k = H^T(x^k) W^{-1} [z - h(x^k)],$$
$$x^{k+1} = x^k + \Delta x^k, \tag{9}$$
$$G(x) = \frac{\partial J(x)}{\partial x} = H^T(x) W^{-1} H(x),$$

where $G$ is the gain matrix, and $\Delta x^k$ is the updated state vector to determine a new vector $x^k$. The iterative calculation proceeds until the preset convergence criterion is attained. The prefix threshold $\varepsilon$ is compared to the largest absolute value of the updated state vector ($\Delta x^k$), and the state variables will be determined in the last iteration when $\max(|\Delta x|) < \varepsilon$.

$H(x) = \partial h(x)/\partial(x)$ is the Jacobian matrix, which is derived from the function vector $h(x)$.

$$H(x) = \frac{\partial h(x)}{\partial(x)} = \begin{bmatrix} \frac{\partial h_1(x)}{\partial x_1} & \frac{\partial h_1(x)}{\partial x_2} & \cdots & \frac{\partial h_1(x)}{\partial x_n} \\ \frac{\partial h_2(x)}{\partial x_1} & \frac{\partial h_2(x)}{\partial x_1} & \cdots & \frac{\partial h_2(x)}{\partial x_n} \\ \vdots & \vdots & \ddots & \cdots \\ \frac{\partial h_m(x)}{\partial x_1} & \frac{\partial h_m(x)}{\partial x_1} & \cdots & \frac{\partial h_m(x)}{\partial x_n} \end{bmatrix}. \tag{10}$$

The measurement function definitions and the Jacobian matrix in the SE computations are different when either traditional or smart measurements are included in the measurement

vector [73]. In (11), the Jacobian matrix is shown when traditional ($H_{Traditional}$) and smart ($H_{Smart}$) measurements are included in the measurement vector, respectively.

$$
H_{Traditional} = \begin{bmatrix} \dfrac{\partial V_{SCADA}}{\partial \theta} & \dfrac{\partial V_{SCADA}}{\partial V} \\[2mm] \dfrac{\partial P_{inj}}{\partial \theta} & \dfrac{\partial P_{inj}}{\partial V} \\[2mm] \dfrac{\partial Q_{inj}}{\partial \theta} & \dfrac{\partial Q_{inj}}{\partial V} \\[2mm] \dfrac{\partial P_{flow}}{\partial \theta} & \dfrac{\partial P_{flow}}{\partial V} \\[2mm] \dfrac{\partial Q_{flow}}{\partial \theta} & \dfrac{\partial Q_{flow}}{\partial V} \end{bmatrix} \quad H_{Smart} = \begin{bmatrix} \dfrac{\partial V_{SCADA}}{\partial \theta} & \dfrac{\partial V_{SCADA}}{\partial V} \\[2mm] \dfrac{\partial \theta_{\theta_V}^{PMU}}{\partial \theta} & \dfrac{\partial V_{PMU}}{\partial V} \\[2mm] \dfrac{\partial \theta_{\theta_I}^{PMU}}{\partial \theta} & \dfrac{\partial I_{PMU}}{\partial V} \\[2mm] \dfrac{\partial I_{imag}}{\partial \theta} & \dfrac{\partial I_{imag}}{\partial V} \\[2mm] \dfrac{\partial P_{inj}}{\partial \theta} & \dfrac{\partial P_{inj}}{\partial V} \\[2mm] \dfrac{\partial Q_{inj}}{\partial \theta} & \dfrac{\partial Q_{inj}}{\partial V} \\[2mm] \dfrac{\partial P_{flow}}{\partial \theta} & \dfrac{\partial P_{flow}}{\partial V} \\[2mm] \dfrac{\partial Q_{flow}}{\partial \theta} & \dfrac{\partial Q_{flow}}{\partial V} \end{bmatrix} \tag{11}
$$

## 4. Methodology

As mentioned earlier, SCADA measurements are commonly available from the beginning operation of ADNs to perform DSSE calculations and to make them observable [74]. The installation of PMU technology has gradually been developed as parts of grid modernization projects move from traditional DNs to smart DNs [75]. Thus, DSSE results will be improved by incorporating PMU measurements as well as SCADA measurements [67]. Unfortunately, SCADA measurements typically have a slower refresh rate compared with the newly installed PMU measurements [76]. With smart DNs, it is important to perform DSSE calculations as frequently as possible in order to understand the dynamic state of the system [77]. Rather than only performing SEs as each new SCADA measurement arrives, we want to perform DSSE calculations with every new PMU measurement. This requires a method for interpolating SCADA measurements so they can be incorporated into the DSSE algorithm. In this paper, we assume that we start with an ADN with SCADA measurements. After a period of time, the ADN is updated to include PMUs as part of a grid modernization procedure.

Figures 2 and 3 illustrate how we trained a regression model to predict future SCADA values prior to and after installing PMUs. Regression models are a category of supervised machine learning, where the corresponding output ($y$) at each input ($x$) is available [78]. In the training phase, a relationship between $x$ and $y$ is determined and expressed as $f$; i.e., $y = f(x)$. A regression model was chosen in this study, since there were enough data points to train a model. Note that the regression model was only used for DSSE calculations after PMUs were installed in order to produce interpolated SCADA measurements that could be used to perform DSSE calculations more frequently. In this work, we pre-trained a SCADA prediction model prior to PMU installation and regularly updated the model after PMU installation.

Prior to PMU installation, DSSE calculations based on WLS method were performed using SCADA and pseudo-measurements. A multi-output regression model was trained based on these DSSE results to predict an interpolated SCADA measurement, as shown in Figure 2. The ground truth used for the interpolated SCADA measurement was obtained using backward–forward power flow calculations. This procedure was repeated for every SCADA measurement that arrived prior to the PMU installation.

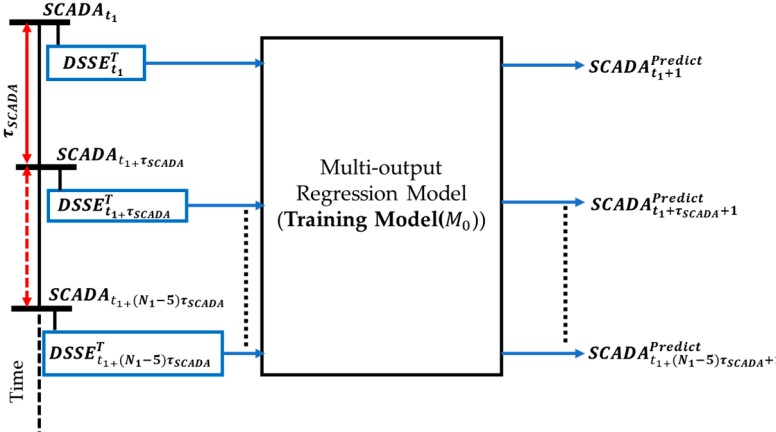

**Figure 2.** A multi-output regression model to predict SCADA measurements. For the figure, we assume that the PMUs are installed in the DN at time step $N_1$, so the procedure is performed through time step $N_1$-5.

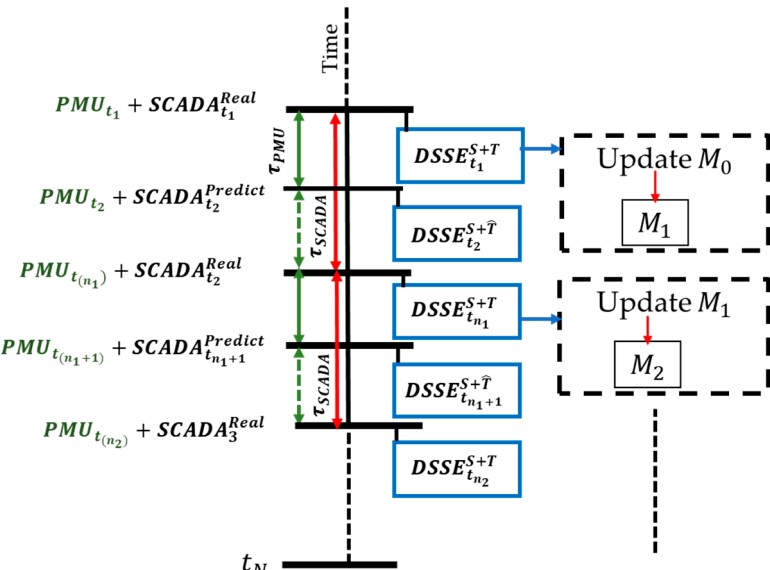

**Figure 3.** DSSE calculations and updating a training model after PMU installation.

A different training procedure was performed after adding the PMUs to the ADN. In the updated procedure, SCADA measurements were predicted using the most recent DSSE calculations, which in turn depended on the most recent PMU measurement and the most recent SCADA measurement or prediction. As shown in Figure 3, a training model was updated every time real SCADA measurements were provided in order to improve model performance and make it robust against dynamic changes of ADNs. For instance, when SCADA and PMU measurements were refreshed at $t_1$, a training model was updated from $M_0$ to $M_1$.

In order to evaluate the DSSE result, the Mean Absolute Error (MAE) and Root Mean Square Error (RMSE) are calculated using (12) and (13), respectively:

$$\text{MAE} = \frac{1}{N} \sum_{i=1}^{N} \left| \frac{\hat{x}_i - x_i}{x_i} \right| \tag{12}$$

$$\text{RMSE} = \sqrt{\frac{\sum_{i=1}^{N} (\hat{x}_i - x_i)^2}{N}}. \tag{13}$$

In the above Equations, $\hat{x}$ is the estimated value, $x$ is the actual value, and $n$ refers to the dataset size.

The interface between MATLAB (2022) and Anaconda Python was used to train and update a model using the DSSE results obtained from the SCADA and PMU measurements. All tests were performed on a desktop computer with an Intel Core i5 processor clocked at 1.60 GHz with 8.00 GB RAM.

## 5. Results

### 5.1. Case Study I

The effectiveness of the proposed method was evaluated on the modified IEEE 33 bus distribution network as one of the case studies (shown in Figure 4). The actual values of a DN were calculated based on power flow calculations.

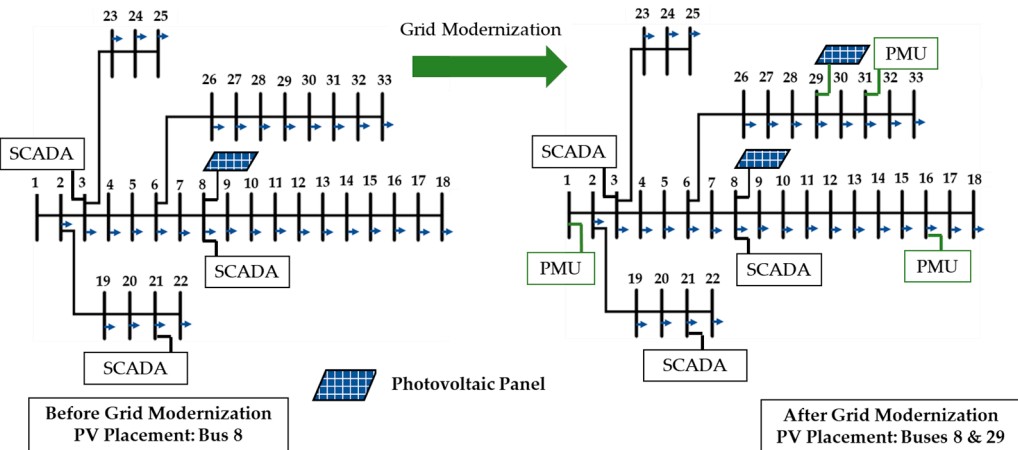

**Figure 4.** IEEE standard 33-bus distribution system before and after grid modernization. Prior to modernization, there was a PV on bus 8 and SCADA units on buses 3, 8, and 21. After modernization, an additional PV was installed on bus 29, and PMU devices were installed on buses 1, 16, and 31.

The following assumptions were made about the system **prior to PMU installation**:

1. Measurements were randomly calculated based on their probability density function for each Monte Carlo trial.
2. Gaussian distribution, with $3\sigma = 50\%$ of the nominal value, was considered for the power injection on the buses [79].
3. We assumed that one Photovoltaic (PV) panel was placed on bus 8 [80] with a maximum generation of 104 kW (The PV was modeled as a negative load), and three SCADA devices were located at bused 3, 8, and 21. Voltage magnitudes, active and reactive power flows, and injections were considered as real measurements ($|V|_3, |V|_8, |V|_{21}$), ($P^{flow}_{3-4}, P^{flow}_{3-23}, P^{flow}_{8-9}, P^{flow}_{21-22}$), ($Q^{flow}_{3-4}, Q^{flow}_{3-23}, Q^{flow}_{8-9}, Q^{flow}_{21-22}$), ($P^{inj}_3, P^{inj}_8, P^{inj}_{21}$), and ($Q^{inj}_3, Q^{inj}_8, Q^{inj}_{21}$) from SCADA measurements to perform DSSE calculations.
4. The pseudo-measurements of the active and reactive power injections and flows were generated to make the system observable and to perform WLS calculations.
5. The standard deviation was considered as 50% of the nominal value for pseudo-measurements, 2% of the voltage magnitude, and 3% of the actual value of active and reactive power flow and injection measurements [44].
6. The voltage magnitudes and phase angles for all the buses were considered as a state variable: $x = [\delta_2, \cdots, \delta_N, V_1, \cdots, V_N]$, where $\delta_N, V_N$ are the voltage phase angle and magnitude, respectively, and $N$ is the number of buses. It was assumed that there were no measurement devices installed in the slack bus. As well, $\delta_1 = 0$ and $V_1 = 1$. The Jacobian matrix will be formed as $H_{Traditional}$, which is shown in Section 3.
7. SCADA measurements were refreshed every five iterations, and DSSE calculations were performed using SCADA and pseudo-measurements every five iterations, con-

cluding at $N_{MC} = 5995$. After this, Monte Carlo trial and PMUs were added to the system.

8.  The DSSE results would be used for training a model to interpolate the SCADA measurements, as discussed in Section 4. It should be noted that the inputs were corrupted by random Gaussian error, with $3\sigma = 2\%$ and $3\sigma = 1$ *crad* for the voltage magnitude and phase angle values, respectively, before feeding to a training model. This initial model would be used to predict the SCADA measurements after the installation of the PMUs to enable the DSSE calculations to be performed more frequently.

9.  WLS calculations were performed using MATLAB, and the iteration process was stopped when the minimum difference between two iterations was $\varepsilon = 10^{-6}$.

The following assumptions were made about the system after **PMU installation**:

1.  Three PMUs were placed at buses 1, 16, and 31 [80], and one more PV with a maximum generation of 80 kW was added to a DN at bus 29 at $N_{MC} = 6000$ to meet the new demands of the DNs in grid modernization development.

2.  To model the uncertainty of the PMU measurements, a Gaussian error with $3\sigma = 1\%$ and $3\sigma = 1$ *crad* was added to the voltage and branch current magnitudes, as well as the voltage and branch current phase angles, respectively [81].

3.  Since a PMU was installed in the slack bus, the phase angle at bus 1 was included in the state vector, so the full state vector is defined as: $x = [\delta_1, \cdots, \delta_N, V_1, \cdots, V_N]$.

4.  We assumed that the PMU measurements were updated at each iteration, i.e., $\tau_{SCADA} = 5\tau_{PMU}$, and, at each iteration, the SCADA measurements were predicted from a training model.

5.  The training model was updated every time N_MC was a multiple of 20 (6020, 6040, 6060, etc.) using the DSSE results obtained from the SCADA measurements and PMU measurements during that time.

6.  In order to verify the proposed method performance, the DSSE results were compared to when the SCADA measurements were replaced using a sample-and-hold technique.

7.  The estimated value from WLS calculations and the actual value from power flow calculations were compared using MAE and RMSE criteria and shown in Table 1.

**Table 1.** DSSE results of voltage magnitudes and phase angles after grid modernization.

| | Voltage Magnitude (*p.u.*) | | Phase Angle (*rad*) | |
|---|---|---|---|---|
| Criteria | Proposed Method | Sample-and-hold technique | Proposed Method | Sample-and-hold technique |
| MAE | $8.2 \times 10^{-4}$ | $9.7 \times 10^{-4}$ | 0.0060 | 0.0071 |
| RMSE | $1.02 \times 10^{-5}$ | $1.45 \times 10^{-5}$ | $7.9 \times 10^{-5}$ | $9.2 \times 10^{-5}$ |

It is clear from Table 1 that the MAE and RMSE values for the state variables (voltage magnitude and phase angle) were improved from the proposed method compared to the sample-and-hold technique. This means that predicting the SCADA measurements based on updating a training model has a better performance. This was true even after more PVs were introduced to the DN, thus showing that our method is robust in a more dynamic environment.

### 5.2. Case Study II

The effectiveness of the proposed method was also evaluated using the modified IEEE standard 69-bus distribution network (shown in Figure 5). The system is suitably adopted to include a mix of commercial and residential loads and DGs. A set of experimental data (available for a time period of one year), obtained from Open Energy Information (OpenEI) [82], was utilized with the simulation time step of 1 h. The hourly data of power generation of a photovoltaic system were computed and adopted based on the actual

data for Bozeman, MT, USA [83,84]. Because reactive power injection is not available, the reactive power injection at bus *i* with a random power factor is defined as [85]:

$$Q_i(\text{t}) = P_i(t) \frac{\sqrt{1 - Pf_i^2(t)}}{Pf_i(t)}, \tag{14}$$

where $Pf_i(t) \sim Unif(0.85, 0.95)$.

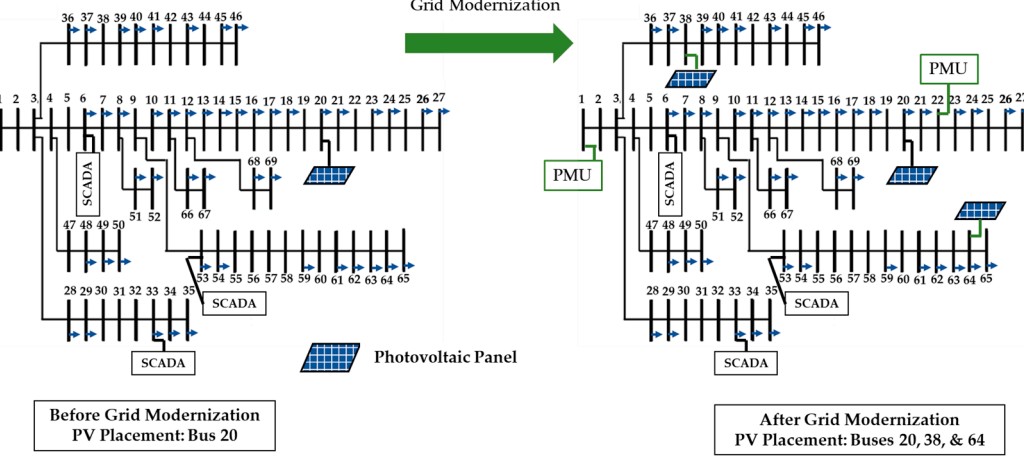

**Figure 5.** IEEE standard 69-bus distribution system before and after grid modernization. Prior to modernization, there was a PV on bus 20 and SCADA units on buses 33 and 53. After modernization, two additional PVs were installed on bus 38 and 64, and PMU devices were installed on buses 1 and 22.

The following assumptions were made about the system **prior to PMU installation**:

1. We assumed that one Photovoltaic (PV) panel was placed on bus 20 [80] with a maximum generation of 15.33 kW (The PV was modeled as a negative load), and three SCADA devices were located at bused 6, 33, and 53. Voltage magnitudes, active and reactive power flows, and injections were considered as real measurements ($|V|_6, |V|_{33}, |V|_{53}$), ($P_{6-7}^{flow}, P_{33-34}^{flow}, P_{53-54}^{flow}$), ($Q_{6-7}^{flow}, Q_{33-34}^{flow}, Q_{53-54}^{flow}$), ($P_6^{inj}, P_{33}^{inj}, P_{53}^{inj}$), and ($Q_6^{inj}, Q_{33}^{inj}, Q_{53}^{inj}$) from SCADA measurements to perform DSSE calculations.
2. The pseudo-measurements of the active and reactive power injections and flows were generated to make the system observable and to perform WLS calculations.
3. The standard deviation was considered as 50% of the nominal value for pseudo-measurements, 2% of the voltage magnitude, and 3% of the actual value of the active and reactive power flow and injection measurements [44].
4. The voltage magnitudes and phase angles for all the buses were considered as a state variable: $x = [\delta_2, \cdots, \delta_N, V_1, \cdots, V_N]$, where $\delta_N, V_N$ are the voltage phase angle and magnitude, respectively, and $N$ is the number of buses. It was assumed that there were no measurement devices installed in the slack bus. As well, $\delta_1 = 0$ and $V_1 = 1$. The Jacobian matrix will be formed as $H_{Traditional}$, which is shown in Section 3.
5. The SCADA measurements were refreshed every five iterations, so DSSE calculations were performed using SCADA and pseudo-measurements every five iterations, concluding at $t = 4125$. (After this time, PMUs were added to the system.)
6. The DSSE results would be used for training a model to interpolate the SCADA measurements, as discussed in Section 4. This initial model would be used to predict the SCADA measurements after the installation of the PMUs to enable DSSE calculations to be performed more frequently.
7. The WLS calculations were performed using MATLAB, and the iteration process was stopped when the minimum difference between two iterations was $\varepsilon = 10^{-6}$.



The following assumptions were made about the system **after PMU installation**:

1. Two PMUs were placed at buses 1 and 22 [80], and two more PVs were added to a DN at buses 38 and 64 with maximum generations of 20.83 kW and 16.67 kW, respectively, at $t = 4130$ to meet the new demands of the DNs in grid modernization development.
2. To model the uncertainty of the PMU measurements, a Gaussian error with $3\sigma = 1\%$ and $3\sigma = 1\ crad$ was added to the voltage and branch current magnitudes, as well as the voltage and branch current phase angles, respectively [81].
3. Since a PMU was installed in the slack bus, the phase angle at bus 1 was included in the state vector and is defined as: $x = [\delta_1, \cdots, \delta_N, V_1, \cdots, V_N]$.
4. We assumed that the PMU measurements were updated at each iteration, i.e., $\tau_{SCADA} = 5\tau_{PMU}$, and, at each iteration, the SCADA measurements were predicted from a training model.
5. The training model was updated once or twice per day using the DSSE results obtained from the SCADA measurements and PMU measurements during that time. The PMU measurement resolution was 20 measurements per day, so the model was updated every 10 (twice daily updates) or 20 (once daily updates) PMU measurements.
6. In order to verify the proposed method performance, the DSSE results were compared when the SCADA measurements were replaced using a sample-and-hold technique.
7. The estimated value from WLS calculations and the actual value from the power flow calculations were compared using MAE and RMSE criteria and shown in Table 2.

**Table 2.** DSSE results of voltage magnitudes and phase angles after grid modernization when the training model was updated every 10 or 20 time steps (twice or once per day).

| | **Voltage Magnitude** (*p.u.*) | | | **Phase Angle** (*rad*) | | |
|---|---|---|---|---|---|---|
| | Proposed Method | | Sample-and-hold technique | Proposed Method | | Sample-and-hold technique |
| Criteria | Daily Update | Twice Daily Update | | Daily Update | Twice Daily Update | |
| MAE | 0.0031 | 0.0023 | 0.0035 | 0.0069 | 0.0062 | 0.0071 |
| RMSE | $4.8 \times 10^{-5}$ | $3.7 \times 10^{-5}$ | $5.2 \times 10^{-5}$ | $9.1 \times 10^{-5}$ | $8.5 \times 10^{-5}$ | $9.8 \times 10^{-5}$ |

This case study was based on actual data, and it had more severe dynamic behaviors. When the model was updated daily, the results were improved, but they were still close to those obtained using the sample-and-hold technique. In order to improve the results, the training model was updated twice a day to become robust against the dynamic behaviors of a DN. As it is clear from Table 2, the MAE and RMSE values for the state variables (voltage magnitude and phase angle) were improved using the proposed method when the model was updated twice a day compared to the sample-and-hold technique.

In Figure 6, the estimated and actual values from the WLS and power flow calculations at bus 20 for 145 consecutive samples and for all buses at a given time are shown.

As it is clear from a Figure 6, the estimated values from the WLS method using SCADA measurements from the proposed method and PMU measurements correctly follow the actual values, which were calculated from the power flow calculations.

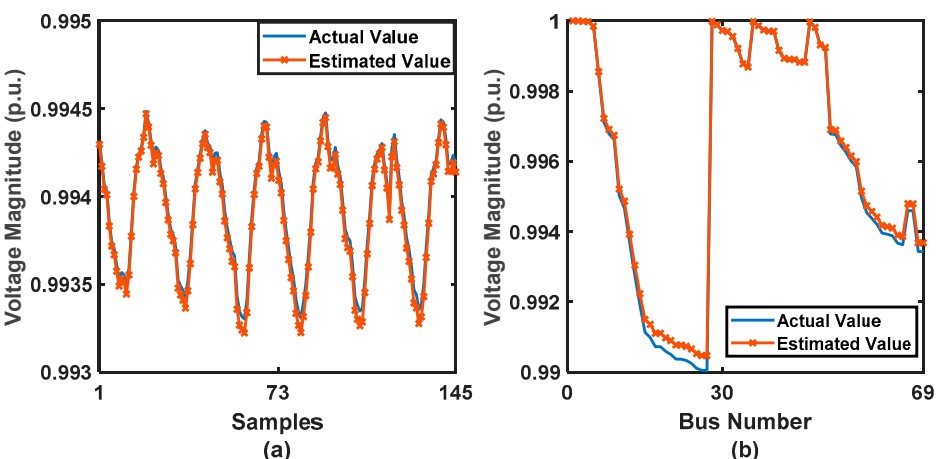

**Figure 6.** (**a**) Actual and estimated values of voltage magnitudes at bus 20 for 145 consecutive samples and (**b**) actual value and estimated values for all buses at a given time.

## 6. Conclusions

In this paper, a new approach based on a regression model was proposed to utilize all the available measurements to improve the DSSE results. The DSSE results have widely been dependent on traditional measurements before the installation of smart measurement devices in ADNs. As part of grid modernization efforts toward an advanced grid, smart measurement devices have progressively been installed to manage and control the dynamic behaviors of ADNs. However, all traditional measurement devices cannot be substituted with advanced ones over a short time, and it is essential to utilize different kinds of measurements using an appropriate method. The method we presented can be used to effectively model distribution systems throughout a grid modernization process. Conventional measurements have low update rates, unlike smart measurement devices, which collect measurements at high frame rates. The difference in frame rates requires interpolation between the traditional measurements to allow for synchronization between the traditional and advanced sensors.

In this study, the WLS method was initially performed using traditional measurements to calculate the state variables, i.e., the bus voltage magnitudes and phase angles. Then, a network configuration was modified toward a modernized DN by installing smart measurement devices to improve the DSSE results and observability analysis. Since traditional and smart measurements have different sampling rates, the DSSE results using a traditional measurement before smart measurement devices were added to the system, and their data were used to train a machine-learning-based regression approach. This training model was used to predict the traditional measurements when these measurements were not available between their refreshing times after the installation of smart measurement devices.

The proposed regression method works effectively for DSSE calculations, but it requires the model to be updated regularly in order to be robust in a dynamic environment. In this work, we compared the results of the proposed method to the results obtained when traditional measurements were incorporated into the DSSE calculation using a sample-and-hold technique after smart measurement installation. The effectiveness of the proposed method was validated using two case studies in the presence of DGs.

The DSSE results given in the paper illustrate that the proposed method is better than the sample-and-hold method after advanced measurement installation. The proposed method could be used in advancing distribution grid modernization to enhance DN performance and capabilities.

In future work, we aim to focus on other aspects of grid modernization challenges, such as how to identify and correct for False Data Injection Attacks (FDIAs) on PMU or SCADA measurements.

**Author Contributions:** Conceptualization, S.R. and T.V.; methodology, S.R. and T.V.; software, S.R., T.V. and K.L.; validation, T.V., K.L. and B.M.W.; writing—original draft preparation, S.R.; writing—review and editing, T.V., B.M.W. and H.N.; visualization, S.R.; supervision, B.M.W.; project administration, H.N.; funding acquisition, H.N. All authors have read and agreed to the published version of the manuscript.

**Funding:** This work was partially supported by the US National Science Foundation under Award 1806184 and by Montana State University.

**Institutional Review Board Statement:** Not applicable.

**Informed Consent Statement:** Not applicable.

**Data Availability Statement:** Not applicable.

**Conflicts of Interest:** The authors declare no conflict of interest.

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
