# Peer review of "Distribution System State Estimation Using Hybrid Traditional and Advanced Measurements for Grid Modernization"

_applsci, doi:10.3390/app13126938_

Round 1

Reviewer 1 Report

This paper provides a comprehensive study on Distribution System State Estimation Using Hybrid Traditional and Advanced Measurements for Grid Modernization.

Overall, the paper presents a well-organized of the current state of research. However, there are several areas where the paper can be improved.

Suggestions to improve the paper:

1. Clarify the research gap: The paper should clearly state the research gap that the study aims to fill. This will help readers understand the importance of the study and its contribution to the existing literature.

2. Improve the introduction: The introduction should provide a clear and concise overview of the paper, including the objectives, research questions, and methodology.

3. Provide more details on the methodology: The paper should provide more details on the methodology used for the review. This will help readers understand how the review was conducted and how the papers were selected.

4. Discuss the limitations of the study: The paper should discuss the limitations of the study and the implications for future research.

5. Provide more details on the simulation results: The paper should provide more details on the simulation results, including the parameters used in the simulations and the software used for the simulations.

6. Provide more discussion on the implications of the findings: The paper should provide more discussion on the implications of the findings.

7. Provide more references: The paper should provide more references to support the claims made in the paper.

8. Improve the language and writing style: The paper should be revised to improve the language and writing style to make it more clear and concise.

9. Provide more details on the experimental setup if any The paper should provide more details on the experimental setup, including the equipment used and the conditions under which the experiments were conducted.

10. Discuss the impact of the proposed technique on the cost of wind turbine systems.

Overall, the paper provides a valuable contribution, and paper can be improved by addressing the above suggestions.

Author Response

Reviewer 1 

Suggestions to improve the paper:

1. Clarify the research gap: The paper should clearly state the research gap that the study aims to fill. This will help readers understand the importance of the study and its contribution to the existing literature.

Response 1: We thank the reviewer for this comment. We address this point by adding two subtitles to introduction section. Moreover, we added more references and explanations to emphasize the importance of the proposed method.

  • Literature Review
  • Contribution

In [49], a spatio-temporal estimation generative adversarial network (ST-EGAN) is proposed to generate pseudo-measurements to perform DSSE calculations in high resolution when SCADA and PMU measurements are not available simultaneously. In [50], an improved sequential state estimation method is proposed to take advantage of AMI and ( PMUs) measurements to address the asynchronization issue for performing DSEE calculations.

As a result, the system cannot be effectively modeled the moment smart measurements are available. In addition, any model training must be redone every time a new sensor is added to the network.

2. Improve the introduction: The introduction should provide a clear and concise overview of the paper, including the objectives, research questions, and methodology.

Response 2: We thank the reviewer for this comment. We address this point by adding two subtitles to introduction section. Moreover, we added more references and explanations to emphasize the importance of the proposed method.

  • Literature Review
  • Contribution

In [49], a spatio-temporal estimation generative adversarial network (ST-EGAN) is proposed to generate pseudo-measurements to perform DSSE calculations in high resolution when SCADA and PMU measurements are not available simultaneously. In [50], an improved sequential state estimation method is proposed to take advantage of AMI and ( PMUs) measurements to address the asynchronization issue for performing DSEE calculations.

As a result, the system cannot be effectively modeled the moment smart measurements are available. In addition, any model training must be redone every time a new sensor is added to the network.

3. Provide more details on the methodology: The paper should provide more details on the methodology used for the review. This will help readers understand how the review was conducted and how the papers were selected.

Response 3: We thank the reviewer for this comment. We address this point by adding more about simulation environment to the methodology section.

The interface between MATLAB (2022) and Anaconda-Python is used to train and update a model using the DSSE results obtained from the SCADA and PMU measurements. All tests were performed on a desktop computer with an Intel Core i5 processor clocked at 1.60 GHz with 8.00 GB RAM.

4. Discuss the limitations of the study: The paper should discuss the limitations of the study and the implications for future research

Response 4: We thank the reviewer for this comment. We address this point by adding more about the main goal and limitation of the study to subsection 1.2.

It is worthwhile to mention that in this work, we aim to address the absence of SCADA measurements between PMU measurements using a regression model instead of switching between two different DSSE methods. The regression model is updated as more DSSE results are obtained to improve its accuracy. Note that we do not consider execution time in this study.

5. Provide more details on the simulation results: The paper should provide more details on the simulation results, including the parameters used in the simulations and the software used for the simulations.

Response 5: We thank the reviewer for this comment. We address this point by adding more to assumptions for case studies and methodology section.

WLS calculations were performed using MATLAB and the iteration process is stopped when the minimum difference between two iterations is .

Voltage magnitude and phase angle at each bus for every case study are considered as parameters to evaluate the proposed method. 

The interface between MATLAB (2022) and Anaconda-Python is used to train and update a model using the DSSE results obtained from the SCADA and PMU measurements. All tests were performed on a desktop computer with an Intel Core i5 processor clocked at 1.60 GHz with 8.00 GB RAM.

We added more figures and explanation to verify the proposed method.

(a) Actual and Estimated Values of Voltage Magnitudes at Bus 20 for 145 consecutive Samples, and (b) Actual Value and Estimated Values for all buses at a given time.

6. Provide more discussion on the implications of the findings: The paper should provide more discussion on the implications of the findings.

Response 6: We thank the reviewer for this comment. We address this point by adding more explanation to conclusion part.

The method we present can be used to effectively model distribution systems throughout a grid modernization process.

7. Provide more references: The paper should provide more references to support the claims made in the paper.

Response 7: We thank the reviewer for this comment. We address this point by adding more references to emphasize the importance of the proposed method.

In [49], a spatio-temporal estimation generative adversarial network (ST-EGAN) is proposed to generate pseudo-measurements to perform DSSE calculations in high resolution when SCADA and PMU measurements are not available simultaneously. In [50], an improved sequential state estimation method is proposed to take advantage of AMI and (PMUs) measurements to address the asynchronization issue for performing DSEE calculations.

8. Improve the language and writing style: The paper should be revised to improve the language and writing style to make it more clear and concise.

Response 8: We are unclear as to what the reviewer had in mind when making this comment. We feel the language and organization of the paper are fairly clear and concise. Perhaps the reviewer can point out a specific example that was unclear or too verbose. A native English speaker has reviewed the manuscript and found the language to be acceptable.

9. Provide more details on the experimental setup if any. The paper should provide more details on the experimental setup, including the equipment used and the conditions under which the experiments were conducted.

Response 9: We thank the reviewer for this comment. We address this point by adding more to assumptions for case studies and methodology section.

WLS calculations were performed using MATLAB and the iteration process is stopped when the minimum difference between two iterations is .

Voltage magnitude and phase angle at each bus for every case study are considered as parameters to evaluate the proposed method. 

The interface between MATLAB (2022) and Anaconda-Python is used to train and update a model using the DSSE results obtained from the SCADA and PMU measurements. All testes are completed in a 1.60 GHz, 8.00 GB of RAM, Intel Core i5.

10.Discuss the impact of the proposed technique on the cost of wind turbine systems.

Response 10: We thank the reviewer for this comment, but the cost of wind turbine systems is outside the scope of this paper since we considered only photovoltaic panels as a Distributed generation for simulations.

Reviewer 2 Report

1) In Figure 5 kindly explain the placement of PV's on different busses or nodes. How have you defined the nodes on which the PVs are to be placed? (Page 10)

2) The reactive power of bus i, what does it mean? is it the reactive power support available on the PV busses??? (Page 10)

3) How long will it take to train the model??

4) what are the actual values and the predicted values? Kindly show their comparison.

5) have you used time series data for the generation and load profiles? if yes then kindly show the load flow values that you have made utilizing the predicted values and compare them with the actual values.

Author Response

1. In Figure 5 kindly explain the placement of PV's on different busses or nodes. How have you defined the nodes on which the PVs are to be placed? (Page 10)

Response 1: We thank the reviewer for this comment. We address this point by adding more explanation to Figures 5 and 6. The placement of PVs are selected based on reference [81] .

 Prior to modernization, there is a PV on bus 8 and SCADA units on buses 3, 8, and 21. After modernization, an additional PV is installed on bus 29 and PMU devices are installed on buses 1, 16, and 31.

Prior to modernization, there is a PV on bus 20 and SCADA units on buses 33, and 53. After modernization, two additional PVs are installed on bus 38 and 64, and PMU devices are installed on buses 1, and 22.

2. The reactive power of bus i, what does it mean? is it the reactive power support available on the PV busses??? (Page 10)

Response 2: We thank the reviewer for this comment. We address this point by adding this sentence:

Because reactive power injection is not available ,the reactive power injection  at bus  with random power factor is defined as [86]:

The PV is modeled as a negative load.

3. How long will it take to train the model??

Response 3: We thank the reviewer for this comment. But execution time was not considered in this paper and we mentioned this point at the end of contribution section.

 Note that we do not consider execution time in this study.

Note:(Just for reviewer): But exaction time for IEEE 33 buses is 23 minutes and for IEEE 69 buses is about 12 minutes.

4. what are the actual values and the predicted values? Kindly show their comparison.

Response 4: We thank the reviewer for this comment. We address this point by adding this sentence to each case study assumptions.

Estimated value from WLS calculations and actual value from power flow calculations are compared in MAE and RMSE criteria.

5. have you used time series data for the generation and load profiles? if yes then kindly show the load flow values that you have made utilizing the predicted values and compare them with the actual values.

Response 5: We thank the reviewer for this comment. We address this point by adding Figure 6 (a) and (b) to verify the proposed method.

In Figure 6, estimated and actual values from WLS and power flow calculations at bus 20 for 145 consecutive samples and for all buses at a given time are shown.  

(a) Actual and Estimated Values of Voltage Magnitudes at Bus 20 for 145 consecutive Samples, and (b) Actual Value and Estimated Values for all buses at a given time.

As it is clear from a Figure 6, the estimated values from WLS method using SCADA measurements from the proposed method and PMU measurements correctly follow the actual values which are calculated from the power flow calculations. 
